# A non-invasive far-red light-induced split-Cre recombinase system for controllable genome engineering in mice

Jiali Wu [1,2], Meiyan Wang[1,2], Xueping Yang[1,2], Chengwei Yi [1], Jian Jiang [1], Yuanhuan Yu [1] & Haifeng Ye [1✉]

The Cre-*loxP* recombination system is a powerful tool for genetic manipulation. However, there are widely recognized limitations with chemically inducible Cre-*loxP* systems, and the UV and blue-light induced systems have phototoxicity and minimal capacity for deep tissue penetration. Here, we develop a far-red light-induced split Cre-*loxP* system (FISC system) based on a bacteriophytochrome optogenetic system and split-Cre recombinase, enabling optogenetical regulation of genome engineering in vivo solely by utilizing a far-red light (FRL). The FISC system exhibits low background and no detectable photocytotoxicity, while offering efficient FRL-induced DNA recombination. Our in vivo studies showcase the strong organ-penetration capacity of FISC system, markedly outperforming two blue-light-based Cre systems for recombination induction in the liver. Demonstrating its strong clinical relevance, we successfully deploy a FISC system using adeno-associated virus (AAV) delivery. Thus, the FISC system expands the optogenetic toolbox for DNA recombination to achieve spatio-temporally controlled, non-invasive genome engineering in living systems.

[1] Synthetic Biology and Biomedical Engineering Laboratory, Biomedical Synthetic Biology Research Center, Shanghai Key Laboratory of Regulatory Biology, Institute of Biomedical Sciences and School of Life Sciences, East China Normal University, Dongchuan Road 500, Shanghai 200241, China. [2] These authors contributed equally: Jiali Wu, Meiyan Wang, Xueping Yang. ✉email: hfye@bio.ecnu.edu.cn

Site-specific DNA recombination is a powerful technique for genome engineering to manipulate transgenes in the chromosomes of both prokaryotic and eukaryotic organisms. Cre recombinase, a site-specific recombinase of the integrase family, catalyzes homologous DNA recombination between two pairs of 34-bp sequences called *loxP* sites[1,2], and can be exploited to induce or silence gene expression for conditional knock-in and knock-out transgenic models. The versatile Cre-*loxP* recombination system has been widely used as a site-specific genetic manipulation tool to precisely manipulate genomes of mammalian cells and transgenic animals in applications such as cell fate mapping[3,4], genome engineering[5–7], and disease treatment[8,9] due to its simplicity and efficiency[1,2]. Previous studies have shown how the basic Cre-*loxP* technology can be combined with chemical-inducible systems such as tetracycline[10,11], tamoxifen[12,13], and rapamycin[14] to achieve temporal control of genome engineering[15]. However, challenges with these chemical-inducible Cre-*loxP* systems include cytotoxicity, leakiness, off-target recombination, as well as limited ability to control systems with high spatiotemporal resolution[16–18].

Moving beyond these constraints with chemical-inducible systems, optogenetics technologies have opened exciting opportunities for studies in neuroscience and many other life science fields, enabling researchers to achieve spatial and temporal control of genes, including applications in gene- and cell-based therapies[19–22]. Compared to chemical agents, light is an excellent inducer for spatiotemporally controlled gene expression. There are light-inducible Cre-*loxP* systems based on UV[23–25], yet these systems can result in cytotoxicity (for example by directly damaging DNA). There are also two blue light-inducible Cre-*loxP* recombination systems, both of which rely on the split-Cre recombinase concept. In the CRY2-CIB1 split-Cre (CRY2-Cre) system, the two Cre fragments component are fused to the blue-light-sensitive plant photoreceptor cryptochrome 2 (CRY2) or its binding domain CIB1[26]. In the PA-Cre system, the two Cre fragments are fused to either positive Magnet (pMag) or negative Magnet (nMag) domains[27]. While these systems have been employed to spatiotemporally control gene expression in vivo, certain limitations are now evident, for example the poor penetrative capacity of blue light through turbid human tissues, and relatively low induction efficiency in living mice, which necessitate long exposure times, thereby increasing phototoxic effects on cells.

Alternative induction energy sources may be one way to help overcome these limitations and develop inducible Cre-*loxP* systems better suited for in vivo and clinical applications. Longer wavelength light sources should be superior inducer energies, as far-red light (FRL; > 700 nm) and near-infrared radiation (NIR; up to 980 nm) are known to penetrate more deeply into living tissues and organs in vivo[28–32]. Although there is no reported inducible Cre system triggered by these lower energy light sources, there are several protein-nanoparticle optogenetic systems responsive to NIR. These systems are based on lanthanide-doped upconversion nanoparticles (UCNPs), which convert radiation from near-infrared lasers (800 or 980 nm) to blue light to activate either the blue-light-responsive channelrhodopsin-2 protein[29] or the light, oxygen, and voltage (LOV2) protein[33]. A notable limitation of these longer wavelength induction methods is the requirement to introduce UCNPs into living systems, which results in cytotoxicity and is a major barrier preventing extensive application in the clinic. Photoactivation of extracellular-signal-regulated kinase (ERK) signaling pathway is accomplished in the mouse auricular epidermis triggered by two-photo excitation (810 and 880 nm), which requires sophisticated hardware[34].

Here, we design and construct a far-red light-induced split Cre-*loxP* system (FISC system) based on split-Cre recombinase and our previously validated FRL-inducible $P_{FRL}$ optogenetic system[35]. In our FISC system, Cre recombinase is split into two fragments, with the N-terminal Cre fragment fused to a Coh2 domain; this CreN-Coh2 fusion protein is constitutively expressed. The C-terminal Cre fragment is fused to a DocS domain, and the inducible expression of this DocS-CreC fusion protein is driven by the FRL-responsive promoter $P_{FRLx}$. Upon illumination with FRL, the activities of Cre recombinase can be reconstituted when the two fragments are brought together based on the strong affinity of the Coh2 and DocS domains[36]. This genetically encoded FISC system, which requires no chemical or nanoparticle components, exhibits low background leakage, has no obvious cytotoxicity effects in mice, and enables high recombination efficiency with precise spatiotemporal control. We demonstrate that the FISC system can be used to induce efficient DNA recombination in vivo, in the internal organs of living mice, where it substantially outperforms the two aforementioned blue-light-based systems for inducing DNA recombination. Finally, we show how the FISC system components can be delivered to mice organs via adeno-associated virus (AAV) vectors, followed by successful induction of DNA recombination upon illumination with a noninvasive FRL LED light. Thus, our study illustrates a powerful tool to optogenetically manipulate genome engineering, enabling a previously unattainable level of noninvasive control that should facilitate studies and therapies focused on a broad range of biological processes.

## Results

**Development of a FRL-induced split Cre-*loxP* system.** Seeking to develop an improved photoactivatable Cre-*loxP* system with deep tissue penetrative capacity and with negligible phototoxicity, we have developed a FRL-induced Cre-*loxP* system by extending our previous bacteriophytochrome-based optogenetic system[35], in which the bacterial photoreceptor BphS can convert intracellular guanylate triphosphate into cyclic diguanylate monophosphate (c-di-GMP) when exposed to FRL[37]. The c-di-GMP then triggers dimerization of the mammalian far-red light-dependent transactivator (FRTA, p65-VP64-BldD), which can then bind to its cognate synthetic light-responsive promoter $P_{FRLx}$ to drive transgene expression. Based on structure analysis of the Cre recombinase[27], we constructed a far-red light-induced split Cre-*loxP* system (FISC system, Fig. 1a) by splitting the Cre recombinase at previously validated residues CreN59/CreC60[27]. Moreover, according to the previous study, the active site of Cre recombinase is located in the larger C-terminal domain[1]. Therefore, in our design, the N-terminal Cre fragment was fused to a Coh2 (CreN59-Coh2) domain, and the fusion protein is constitutively driven by $P_{hCMV}$. The C-terminal Cre fragment was fused to a DocS (DocS-CreC60) domain and this fusion can be inducibly expressed by the FRL-responsive promoter $P_{FRLx}$. Under FRL illumination, the fused fragment DocS-CreC60 is expressed, which restores Cre enzyme catalytic activity by allowing heterodimerization of the Cre-fragment-fused Coh2 and DocS interaction domains. The reconstituted Cre can then excise DNA sequences flanked by two *loxP* sites (Fig.1b).

We initially used human embryonic kidney cells (HEK-293) to test recombination efficiency of the CreN59/CreC60 pair upon FRL illumination, and found that this combination had high induction levels [as assessed by profiling secreted human placental alkaline phosphatase (SEAP) expression] upon FRL illumination; however, it also had strong background activity in dark conditions (Supplementary Fig. 1). To reduce the background activity, we therefore constructed three different FRL-responsive chimeric promoter variants ($P_{FRLx}$) (Fig.1c), and tested different promoter configurations for driving DocS-CreC60

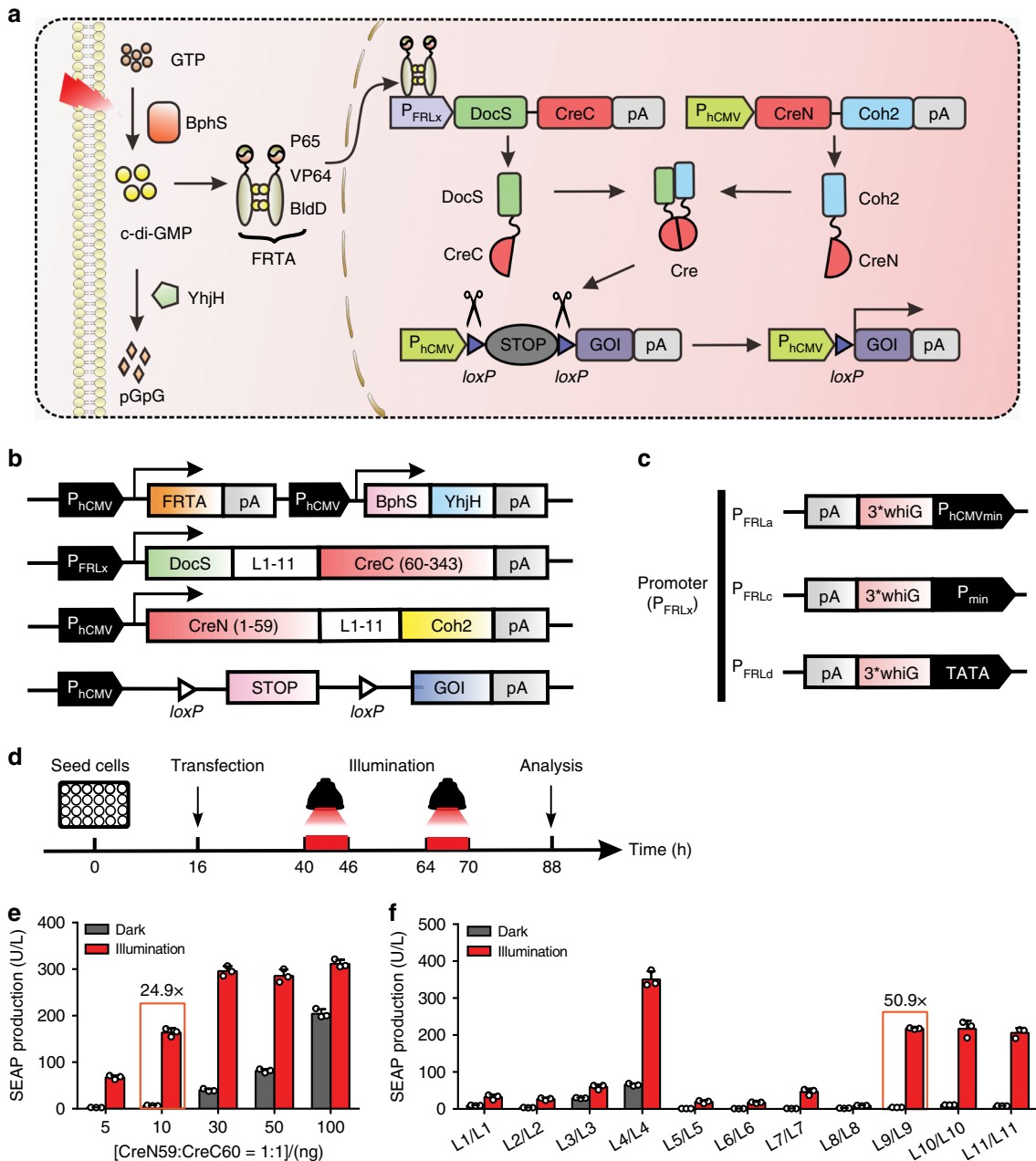

expression and optimized different amounts of the plasmids encoding the CreC60 fusion fragments. Among multiple tested promoters, we found that expressing DocS-CreC60 under the control of the TATA promoter ($P_{FRLd}$) resulted in the light-inducible recombination efficiency in response to FRL illumination (Fig. 1d and Supplementary Fig. 2).

We also found that the highest recombination efficiency was achieved upon cotransfection of HEK-293 cells with four plasmids: the FRL-responsive sensor pXY137 ($P_{hCMV}$-p65-VP64-BldD-pA:: $P_{hCMV}$-BphS-P2A-YhjH-pA, 100 ng), the FRL-inducible CreC60 expression vector pXY133 ($P_{FRLd}$-NLS-DocS-L0-CreC60-pA, 10 ng), the CreN59 expression vector pXY110 ($P_{hCMV}$-CreN59-L0-Coh2-NES-pA, 10 ng), and the Cre-dependent SEAP reporter pGY125 ($P_{hCMV}$-loxP-STOP-loxP-SEAP-pA, 200 ng). This combination resulted in a 24.9-fold induction of DNA recombination efficiency upon FRL illumination (Fig. 1e). To obtain robust DNA

recombination efficiency using this system, we next designed different linkers and optimized the linker amino acid sequences between CreN59 and Coh2 as well as between CreC60 and DocS to avoid a major steric hindrance that would impede the reconstitution of an active structure[14,27]. We compared different combinations of these linkers and found that one of the combinations (L9/L9) yielded the highest fold induction (50.9-fold induction) upon FRL illumination (Fig. 1f, Supplementary Fig. 3 and Tables 1 and 2).

In addition, we further tested the CreN59/CreC60 pair for their catalytic capacity to excise a reporter sequence flanked by various loxP mutants, including lox2722, lox66, lox71, lox72, JTZ17, JTZ17/JT15, JT15, as well as the original (nonmutant) loxP (Supplementary Fig. 4a). Specifically, we constructed SEAP reporters flanked by various loxP mutants (floxed-STOP SEAP) and found that the CreN59/CreC60 pair showed the highest DNA

**Fig. 1 Design and optimization of the far-red light-induced split Cre-*loxP* system (FISC system). a** Schematic representation of the FISC system. Cre recombinase was split into two fragments: CreN59 (residues 1–59) fused to Coh2 driven by a constitutive promoter ($P_{hCMV}$) and CreC60 (residues 60–343) fused to DocS driven by the far-red light (FRL, 730 nm)-inducible promoter ($P_{FRLx}$). Upon FRL illumination, the photoreceptor BphS is activated to convert intracellular guanylate triphosphate (GTP) into cyclic diguanylate monophosphate (c-di-GMP). The cytosolic c-di-GMP production induces binding of the far-red light-dependent transactivator FRTA (p65-VP64-BldD) to its synthetic promoter $P_{FRLx}$ to drive DocS-CreC60 expression. Consequently, the catalytic activities of Cre recombinase can be restored once the two Cre fragments assemble based on affinity interactions of their respective Coh2 and DocS fusion domains, enabling to excise DNA sequences flanked by *loxP* sites. **b** Schematic depicting the genetic configuration of constructs used in the FISC system. pA, polyadenylation signals; YhjH, the bacterial c-di-GMP phosphodiesterase; DocS, dockerin S from *C. thermocellum* complexed with Coh2; $L_{1-11}$, different linkers from 1 to 11 (Supplementary Table 1), Coh2, an anchoring protein from *C. thermocellum*; *loxP*, the specific Cre recombinase binding site; STOP, a terminator containing pA to prevent transcription; GOI, gene of interest. **c** Schematic depicting different genetic configurations of the FRL-inducible promoters $P_{FRLx}$. 3*whiG, three copies of BldD-specific binding sequence; $P_{hCMVmin}$, minimal version of $P_{hCMV}$; $P_{min}$, minimal eukaryotic promoter; TATA, minimal eukaryotic promoter with only TATA box. **d** Schematic depicting the time schedule for the FISC experimental procedure with mammalian cells. **e** Optimization of the different transfection amounts for CreN59-Coh2 expression and light-inducible $P_{FRLd}$-driven DocS-CreC60 expression. HEK-293 (6 × $10^4$) cells were cotransfected with pXY137 ($P_{hCMV}$-p65-VP64-BldD-pA::$P_{hCMV}$-BphS-P2A-YhjH-pA, 100 ng), pGY125 ($P_{hCMV}$-*loxP*-STOP-*loxP*-SEAP-pA, 200 ng), pXY110 ($P_{hCMV}$-CreN59-L0-Coh2-NES-pA) and pXY133 ($P_{FRLd}$-NLS-DocS-L0-CreC60-pA) from 5 to 100 ng at a ratio of 1:1 (w/w), and then illuminated for 6 h with FRL (1.5 mW cm$^{-2}$, 730 nm) once each day for 2 days. SEAP expression in the culture supernatants was profiled at 48 h after the first illumination ($n = 3$ independent experiments). **f** Optimization of the different linkers (L1–L11) between the CreN59 and Coh2 domains, as well as the CreC60 and DocS domains. The 6 × $10^4$ HEK-293 cells per well were cotransfected with pXY137 (100 ng), pGY125 (200 ng), and Docs-CreC60 and Coh2-CreN59 with different combinations of the linkers (10 ng/10 ng) (Supplementary Table 2); these were illuminated as described in **e**, followed by SEAP expression in the culture supernatants profiled at 48 h after the first illumination ($n = 3$ independent experiments). The orange frame in (**e**, **f**) marks the best-performing condition. **e**, **f** Data represent the mean ± SD. Source data for this figure are available in the Source data file.

recombination efficiency for the original *loxP*-flanked reporter; a little or no substantial induction of DNA recombination was observed with other *loxP* mutants-flanked reporter sequences (Supplementary Fig. 4b). As a result of these improvements and tests, we have developed an optimized and specific FISC system that affords highly efficient DNA recombination (50.9-fold induction) under FRL illumination.

**DNA recombination performance of the FISC system.** To demonstrate photoactivatable regulation of Cre activity in mammalian cells, HEK-293 cells were cotransfected with plasmids of this FISC system and fluorescent (EGFP) or luciferase reporters flanked by *loxP* sites. These experiments confirmed that cells equipped with our FISC system exhibited high levels of recombination upon FRL illumination while displaying low background recombination levels for cells kept in the dark (Fig. 2a, b and Supplementary Fig. 5). We next characterized the kinetics of FISC-mediated DNA recombination, and found that recombination upon FRL induction is both illumination-intensity and exposure-time dependent (Fig. 2c, d) without significant background increase over time (Supplementary Fig. 6). We also confirmed that the FISC system can activate DNA recombination in a variety of mammalian cell lines with different performance (Fig. 2e). The variable DNA recombination efficiency between different cell lines can be attributed to cell specific factors such as differences in transfection efficiency and potential interactions with endogenous cell components as also observed in previous studies[35,38,39]. Moreover, experiments using a photomask to restrict the spatial pattern of FRL exposure revealed the high spatial resolution performance of the FISC system (Fig. 2f).

We next evaluated the impact of FRL (730 nm) or blue light (460 nm) illumination on the viability of mammalian cells. HEK-293 cells were transfected with pSEAP2-control and then illuminated with FRL or blue light for different time. The SEAP expression showed that the FRL exposure resulted in negligible cytotoxicity. However, a marked difference was observed from the blue light treatment, which significantly reduced cell viability (Supplementary Fig. 7). Moreover, we did not observe substantially increased cytotoxicity with extended FRL exposure of cells equipped with the FISC system (Supplementary Fig. 8), indicating the inertness of the system constituents. In addition, we compared the DNA recombination efficiency between our FISC system and two existing blue light-controlled Cre-*loxP* recombination systems, including the CRY2-Cre[26] and the PA-Cre system[27] (Supplementary Fig. 9). Our FISC system yields more efficient DNA recombination (46.7-fold induction) than either the CRY2-Cre system (11.6-fold induction) or the PA-Cre system (8.7-fold induction). These in vitro results demonstrating dramatically increased DNA recombination efficiency, viewed alongside our observations of reduced cytotoxicity of FRL compared with blue light exposure, together emphasize that our FISC system represents an attractive approach for light-induced Cre-*loxP* recombination.

**In vivo DNA recombination with FISC system in BALB/c mice.** We next tested whether the FISC system can induce DNA recombination in vivo upon FRL illumination. Prior to infecting/ injecting mice, and seeking a relatively simple system comprising few constructs to facilitate in vivo FISC system delivery, we developed several iterations of a single plasmid which concatenated the constructs for both halves of the split-Cre product. To overcome issues with leaky expression in the dark and reduced induction capacity, we optimized the distance between the promoters for the two split-Cre sequences (Supplementary Fig. 10). We eventually obtained an optimized concatenated split-Cre vector pXY237 (pA-CreC60-L9-DocS-NLS-$P_{FRLd}$-Space3-$P_{hCMV}$-CreN59-L9-Coh2-NES-P2A-ZeoR-pA), which successfully combined the constructs for the two Cre fusion fragments in a way that preserved the design-necessitated constitutive (CreN59-Coh2) and inducible (DocS-CreC60) expression patterns.

We then used hydrodynamic injection (tail vein) to transiently transfect wild-type BALB/c mice with an iteration of the FISC system comprising three plasmids: the FRL-responsive sensor pXY137 (150 μg), the concatenated split-Cre vector pXY237 (100 μg) and the Cre-inducible luciferase reporter pXY185 ($P_{hCMV}$-*loxP*-STOP-*loxP*-Luciferase-pA, 100 μg). Eight hours after injection, the abdomens of the transfected mice were exposed to FRL (20 mW cm$^{-2}$, 730 nm). Compared with control mice which were injected with only the luciferase reporter plasmid pXY185 or the FISC components but left in the dark, quantitative analysis of bioluminescence imaging data showed that the injected mice that received the 12 h FRL illumination exhibited significantly higher luciferase activity (Fig. 3a, b).

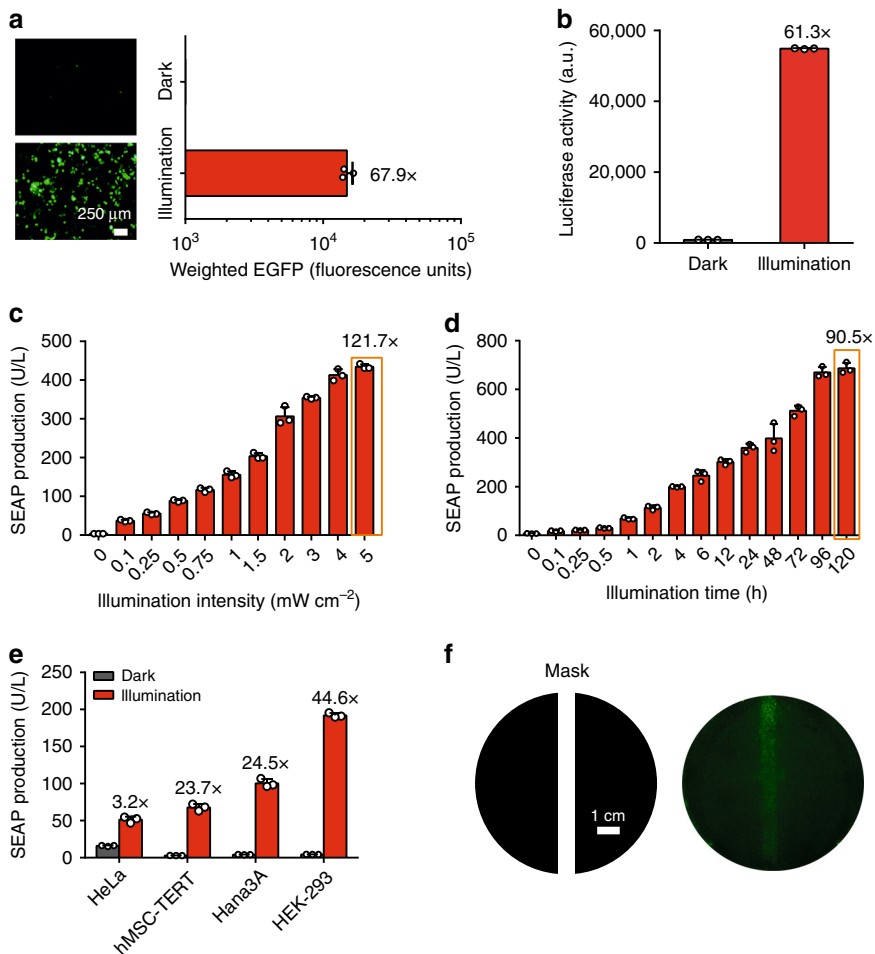

**Fig. 2 Characterization of the FISC system. a** Cre-catalyzed DNA recombination with FISC system using fluorescence imaging and flow cytometry. HEK-293 cells (6 × 10⁴) were cotransfected with pXY137, pXY169 (P_hCMV-CreN59-L9-Coh2-NES-pA), pXY177 (P_FRLd-NLS-DocS-L9-CreC60-pA), and pDL78 (P_hCMV-*loxP*-STOP-*loxP*-EGFP-pA) at a ratio of 10:1:1:20 (w/w/w/w), illuminated for 6 h with FRL (1.5 mW cm⁻², 730 nm) each day for 2 days, and expression of the reporter protein EGFP was visualized by fluorescence microscopy and by flow cytometry at 48 h after the first illumination. Representative images from n = 5 biological replicates. Scale bar, 250 μm. Graph bars represent mean ± SD of n = 3 biological replicates. **b** Cre-catalyzed DNA recombination with FISC system using luciferase assay. HEK-293 cells (6 × 10⁴) were cotransfected with pXY137, pXY169, pXY177, and pXY185 (P_hCMV-*loxP*-STOP-*loxP*-Luciferase-pA) at a ratio of 10:1:1:20 (w/w/w/w), illuminated as described in (**a**), and bioluminescence measurements were taken at 48 h after the first illumination (n = 3 independent experiments). **c** Assessment of illumination-intensity-dependent FISC system activity. 6 × 10⁴ HEK-293 cells were cotransfected with pXY137, pXY169, pXY177, and pGY125 (SEAP reporter plasmid) at a 10:1:1:20 (w/w/w/w) ratio and illuminated with FRL for 6 h each day for 2 days at eight different light intensities (0–5 mW cm⁻²); SEAP levels were profiled at 48 h after the first illumination (n = 3 independent experiments). The orange frame marks the highest-fold induction mediated by FISC system. **d** Exposure-time-dependent FISC system activity. With the same plasmid ratios as in (**c**), we illuminated transfected cells with FRL (1.5 mW cm⁻², 730 nm) for different time periods (0–120 h). SEAP expression was profiled in the cell culture supernatant at 120 h after initial illumination (n = 3 independent experiments). The orange frame marks the highest-fold induction mediated by FISC system. **e** FISC-induced SEAP expression in multiple mammalian cell lines. Four different mammalian cell lines were cotransfected and illuminated as described in (**c**), and SEAP expression in the culture supernatant was profiled at 48 h after the first illumination (n = 3 independent experiments). **f** Evaluation of the spatial resolution for FISC-mediated transgene expression. A monolayer comprising HEK-293 cells was cotransfected with pXY137, pXY169, pXY177, and pDL78 (EGFP reporter plasmid) at a ratio of a 10:1:1:20 (w/w/w/w), and illuminated as described in (**a**), but through a photomask (schematic, left) with a 6.5 mm slit, and fluorescence microscopy based analysis of the corresponding pattern of EGFP expression at 48 h after the first illumination (right). Representative images from n = 2 biological replicates. **b**–**e** Data represent the mean ± SD. Source data for this figure are available in the Source data file.

We also compared the in vivo DNA recombination efficiencies of the FISC system and CRY2-Cre and PA-Cre systems. Mice were illuminated with FRL (20 mW cm⁻², 730 nm) or blue light (20 mW cm⁻², 460 nm) for 12 h (15 min on, 15 min off) starting at 8 h post hydrodynamic injection. Strikingly, illumination of FISC-equipped mice with FRL resulted in a ~54-fold increase in luciferase activity, whereas the maximum increase observed upon blue light illumination of the mice bearing CRY2-Cre or PA-Cre system components was only approximately threefold above the background levels of the dark control animals (Fig. 3b, c). This is

not surprising, given the known capacity of FRL but not blue light energy to penetrate animal tissues. Thus, beyond establishing that our FISC system can efficiently induce DNA recombination in vivo, these results highlight the intrinsic physical properties of FISC system which make it superior to blue-light-based systems for physiological studies at the whole animal level.

**DNA recombination with FISC system in the transgenic mice.** To verify that the FISC system can be used for DNA

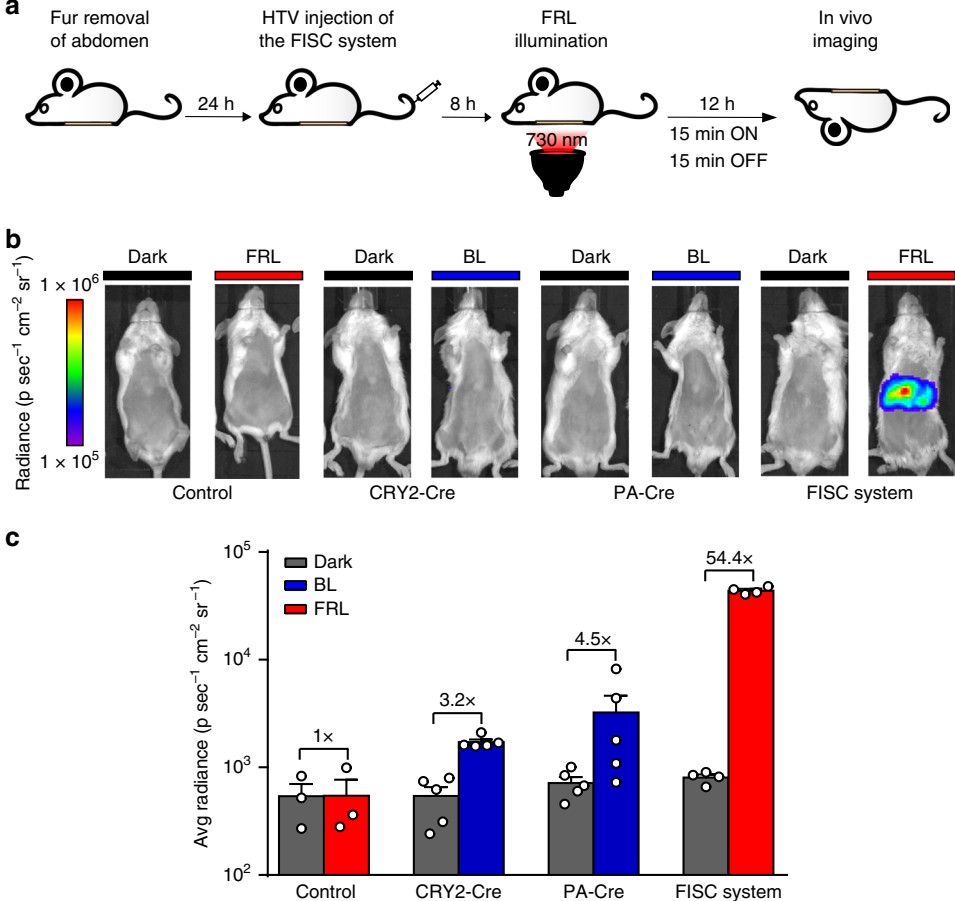

**Fig. 3 FISC-induced DNA recombination in BALB/c wild-type mice. a** Schematic showing the experimental procedure of light-induced DNA recombination activity in mice. **b** Comparison of the FISC system with the CRY2–Cre and PA-Cre systems using in vivo bioluminescence imaging. The BALB/c mice were transiently hydrodynamically injected (tail vein) with an iteration of the FISC system comprising three plasmids [pXY137/pXY237 (pA-CreC60-DocS-NLS-$P_{FRLd}$-Space3-$P_{hCMV}$-CreN59-Coh2-P2A-ZeoR-pA)/pXY185 (luciferase reporter plasmid) at a ratio of 1.5:1:1 (w/w/w)], or the CRY2-Cre, or the PA-Cre system or only the luciferase reporter plasmid pXY185 as control. At 8 h after the injection, the mice were illuminated with FRL (20 mW cm$^{-2}$, 730 nm) or blue light (BL, 20 mW cm$^{-2}$, 460 nm) for 12 h (15 min on, 15 min off, alternating) or maintained in the dark, followed by imaging at 12 h after light illumination. **c** Bioluminescence measurements of the BALB/c mice shown in (**b**) (Control: $n = 3$, FISC system: $n = 4$, CRY2-Cre and PA-Cre: $n = 5$. Data present the mean ± SEM). See Supplementary Table 4 for detailed description of genetic components for each optogenetic system. Source data for this figure are available in the Source data file.

recombination of a genomic locus, we used hydrodynamic injection to introduce a two plasmid iteration of the FISC system into tdTomato transgenic (*Gt(ROSA)26Sor$^{tm14(CAG-tdTomato)Hze}$*) mice, in which a Cre reporter allele was designed to have a *loxP*-flanked-STOP cassette preventing transcription of a CAG promoter-driven red fluorescent protein variant (tdTomato). In the presence of Cre recombinase results in excision of the stop cassette, leading to expression of the reporter tdTomato. Specifically, the pXY137 (150 μg) and pXY237 (100 μg) plasmids were delivered into tdTomato transgenic mice via tail vein injection (Supplementary Fig. 11a), and 8 h after injection, exposed the mice to FRL (20 mW cm$^{-2}$, 730 nm) for 12 h (15 min on, 15 min off, alternating). The isolated livers of the FRL-illuminated mice exhibited intense tdTomato expression signals, such signals were not evident with the FISC-injected dark control mice (Supplementary Fig. 11b–d). In contrast, parallel experiments with the blue light-controlled systems showed only weak induction of tdTomato expression upon blue light illumination (20 mW cm$^{-2}$, 460 nm) (Supplementary Fig. 11b,c). A quantitative analysis showed a significant induction of tdTomato signal intensity in isolated livers from the FRL illuminated mice, whereas no significant inductions were observed for the CRY2-Cre and PA-Cre

mice (Supplementary Fig. 11c). These results support that our FISC system enables DNA recombination of genomic loci in vivo as triggered by noninvasive illumination with an external FRL.

We further confirmed this in vivo FRL-induction DNA recombinase activity of the FISC system in experiments using electroporation of the leg muscles of tdTomato transgenic mice. The tibialis posterior muscles of the mice were electroporated with a total of 40 μg of the plasmids for the FISC system or the blue light-controlled systems, followed by illumination with FRL (20 mW cm$^{-2}$, 730 nm) or blue light (20 mW cm$^{-2}$, 460 nm) for 12 h (15 min on, 15 min off) each day for 2 days (Supplementary Fig. 12a). qRT-PCR and Western blot analyses (Supplementary Fig. 12b,c) both showed that FRL illumination of the FISC-equipped mice induced a substantial increase in the tdTomato signal compared with electroporated dark control mice. No substantial increase in tdTomato signal intensity was observed for either of the blue light-controlled systems in muscle tissues.

**AAV mediated FISC system in tdTomato transgenic mice.** AAVs are often the vector of choice for gene therapies in biological research and in clinical settings, owing to their low immunogenic potential and their reduced oncogenic risk from

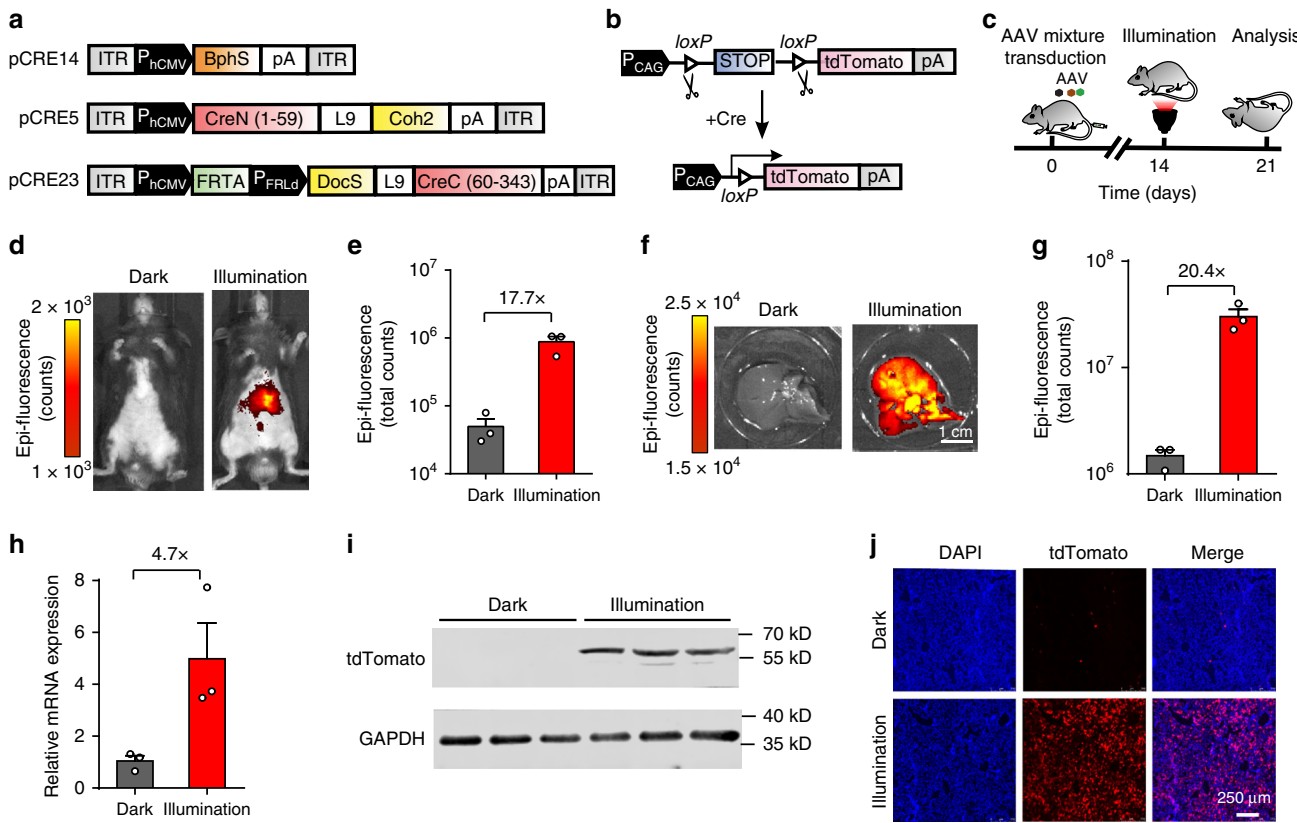

**Fig. 4 AAV delivery of FISC-mediated DNA recombination in transgenic Cre-tdTomato reporter mice. a** Schematic depicting the genetic configuration AAV vectors for the FISC DNA recombination system. **b** Schematic of working principle for transgenic Cre-tdTomato reporter mice in which the *loxP*-flanked STOP cassette can be excised by Cre recombinase to allow Cre reporter (tdTomato) expression. **c** Schematic representation of the experimental procedure for FISC-mediated DNA recombination activity in mice. **d** Representative images of the tdTomato fluorescence in the AAV-transducted reporter mice (*n* = 3 mice/group). **e** The fluorescence measurements of the tdTomato expression shown in (**d**) (*n* = 3 mice/group). **f** Representative images of the tdTomato fluorescence in isolated liver tissue from the AAV-transducted mice shown in (**d**) (*n* = 3 mice/group). Scale bar, 1 cm. **g** Bioluminescence tdTomato expression shown in (**f**) (*n* = 3 mice/group). **h, i** qRT-PCR (**h**) and Western blot (**i**) analysis of tdTomato in isolated liver tissue shown in (**f**) (*n* = 3 mice/group). Western blotting images are representative of three mice from two independent experiments. **j** Representative fluorescence images of the liver sections of the transgenic Cre-tdTomato reporter mice shown in (**f**) (*n* = 5 biological replicates. Scale bar, 250 μm). Data in (**e, g, h**) represent the mean ± SEM. Source data for this figure are available in the Source data file.

host-genome integration[40,41]. Recently, a red-light-activated AAV delivery platform has been reported for optogenetic activation of gene expression in mammalian cells[42]. To enable our FISC system for potential use in clinical therapies for diseases in the future, we verified the ability to deliver our FISC system via an AAV vector into tdTomato transgenic mice. Because of the packaging size limitation of AAV vectors, we deployed our FISC system in three separate viral vectors: the constitutively expressed light-sensing module BphS, the constitutively expressed CreN59-Coh2 fragment, the constitutively expressed hybrid transactivator p65-VP64-BldD and the FRL-induced Docs-CreC60 fragment (Fig. 4a). We then transduced tdTomato transgenic mice a mixture of these three AAVs via tail vein injection (Fig. 4b, c). We assessed whether this AAV-mediated FISC system could be used to induce FRL-dependent in vivo DNA recombination in the tdTomato transgenic mice. At 2 weeks post injection, we illuminated the mice at intensity of 20 mW cm$^{-2}$ for 12 h (15 min on, 15 min off, alternating) each day for 2 days. One week after the initial illumination, fluorescence imaging indicated that mice exposed to FRL exhibited a 17.7-fold increase over transducted dark control mice (Fig. 4d, e); Moreover, we observed higher tdTomato signal in the isolated livers derived from the FRL illuminated mice (Fig. 4f). Analysis of isolated livers revealed a 20.4-fold increase (Fig. 4g) compared with the transducted dark control mice. Similarly, qRT-PCR and Western blot analyses of

isolated liver tissues (Fig. 4h, i) showed that FRL illumination of the mice transducted with the three FISC AAV vectors induced a substantial increase in the tdTomato signal compared with the transducted dark control mice. This FRL-induced increase in tdTomato expression was also confirmed via fluorescence microscopy imaging of liver sections (Fig. 4j). Conclusively, these results demonstrate that AAV-medicated FISC system showed very more efficient performance as compared with the other two delivery methods (hydrodynamic injection and electroporation) in vivo (Supplementary Table 3) which offers a potential approach in clinical therapies in the near future.

## Discussion

Optogenetic tools have revolutionized the study of cell activities and functions, offering previously unimaginable spatiotemporal resolution to researchers[22,34,43–46]. At present, the available light-inducible technologies for the Cre-*loxP* recombination system have been developed based on UV light or on blue light, both of which have very little physical ability to penetrate deeply into tissues[47]. Moreover, such systems are limited by side effects such as phototoxicity, unwanted inflammation, and low recombination efficiency in vivo[27]. In the present study, we demonstrate the successful development of a far-red light-induced split Cre-*loxP* system (FISC system) that can efficiently control DNA

recombination in a light-dependent manner, not only in various mammalian cell lines, but also in mice. Importantly, we show that the FISC system has minimal dark background but has very strongly inducible DNA recombination efficiency (up to 121.7-fold induction in HEK-293 cells upon exposure to FRL). We also demonstrate the light-intensity and light-duration dependent tuning of FISC-mediated DNA recombination activity, and demonstrate that recombination activity can be spatially controlled by for example using a photomask to restrict the area that is exposed to FRL illumination.

It bears emphasis that the FISC system is based on FRL illumination, so it has several advantages over light-inducible gene expression systems that are based on shorter wavelengths of light: given its substantially reduced energy, FRL is less cytotoxic than blue or UV light. Moreover, FRL is able to penetrate deeply into tissues, so it offers researchers (and potentially clinicians) a gene induction and no intracellular signaling perturbation capacities that faced via chemically inducible Cre-loxP recombination systems[14,48–50]. The genetically encoded FISC system presented in this work is very easy to set up and adapted for in vivo applications. Future iterations of the FISC system approach will likely expand beyond the initial proof-of-concept hydrodynamic injection demonstrations, potentially enabling extremely tissue- or even cell-type specific deployment for high resolution studies that will not require invasive or chemical manipulations to achieved conditional expression of genes of interest in animals.

Importantly, our demonstration that the FISC system components can be delivered by AAV to achieve efficient DNA recombination in animals illustrates how our FRL-inducible system is suitable for a potentially numerous of varied applications like creating and conditionally activating knock-in and knock-out organisms with fine spatiotemporal control. Therefore, the FISC system can be applicable for efficient gene manipulation in multiple fields, such as gene deletion in different cell types and organisms for studying gene functions.

Despite the strong DNA recombination efficiency mediated by the FISC system, there is still room for improvement with regard to the convenience or a translational perspective. Because of the packaging size limitation of AAV vectors, we have to deploy our FISC system in three separate AAV viral vectors for in vivo delivery. This is obviously an obstacle for further molecular biology study or translational research in the future. An idea solution is to develop a more packable device with small construct size ensuring efficient delivery in vivo.

Collectively, the described FISC system should provide an improved system allowing precise control of genome engineering in target single cells or whole organisms in a spatiotemporal fashion with the capacity for deep penetration and substantially reduced toxicity and invasiveness. We expect that our FRL-inducible Cre-loxP system will pave the way for the broader applications including optical control in tissue xenografts, the creation of knock-in and knock-out organisms, and the potential to designate stem cell fate and pluripotency induction in a spatiotemporal fashion.

## Methods

**Cloning and plasmid construction**. Plasmids and primers used in this study are provided in Supplementary Tables 4 and 5 and the sequences were confirmed by DNA sequencing (Genewiz Inc., Suzhou, China). The detailed DNA sequence information of the most important genetic modules used in this study was provided in the Supplementary File. Some plasmids were constructed by Gibson assembly according to the manufacturer's instructions (Seamless Assembly Cloning kit; Obio Technology Inc.; Cat. no. BACR(C) 20144001).

**Cell culture and transfection**. Human embryonic kidney cells (HEK-293, ATCC: CRL-1573), human cervical adenocarcinoma cells (HeLa, ATCC: CCL-2), HEK-293-derived Hana3A cells engineered for constitutive expression of RTP1, RTP2,

REEP1, and $G_{\alpha o\lambda \phi}$ and telomerase-immortalized human mesenchymal stem cells (hMSC-TERT) were cultured at 37 °C in a humidified atmosphere containing 5% $CO_2$ in Dulbecco's Modified Eagle Medium (DMEM, Gibco; Cat. no. 31600-083) containing 10% (v/v) fetal bovine serum (FBS, AusGeneX Industries; Cat. no. FBSSA500-S) and 1% (v/v) penicillin/streptomycin solution (Beyotime, China; Cat No. ST488). All the cell lines have been regularly tested for the absence of Myco-plasma and bacterial contamination. All cell lines were transfected with an optimized polyethyleneimine (PEI)-based protocol. Briefly, $6 \times 10^4$ cells were plated per well in a 24-well cell culture plate and cultured for 16 h. The cells were subsequently incubated for 6 h with 50 μL of a 3:1 PEI: DNA mixture (w/w) (poly-ethyleneimine, MW 40,000, stock solution 1 mg mL$^{-1}$ in ddH$_2$O; Polysciences; Cat. no. 24765) containing 0.32 μg of total plasmid DNA for FISC system. Twenty-four hours after the transfection, the cells were illuminated for 6 h each day for 2 days by FRL LED (1.5 mW cm$^{-2}$, 730 nm, Epistar, Taiwan, China). Cell number and viability were determined with a Countess II automated cell counter (Life Technologies).

**Cell viability assay**. The $1 \times 10^4$ HEK-293 cells seeded in a 96-well plate were illuminated with FRL (1.5 mW cm$^{-2}$, 730 nm) or BL (1.5 mW cm$^{-2}$, 460 nm) for 24 h and cell viability was assayed using Cell Counting Kit-8 (Beyotime, China; Cat. no. C0037) according to the manufacturer's protocol. Cell inhibition or pro-liferation was assayed by utilizing highly water-soluble tetrazolium salt, WST-8 [2-(2-methoxy-4-nitrophenyl)-3-(4-nitrophenyl)-5-(2,4-disulfophenyl)-2H-tetra-zolium, a monosodium salt], which produces a water-soluble formazan dye upon reduction in the presence of an electron mediator. After incubation with 10 μL CCK8 at 37 °C for 2 h, the plate was read with a Synergy H1 microplate reader (BioTek Instruments Inc.) at 450 nm.

**DNA recombination performance of the FISC in mammalian cells**. The $6 \times 10^4$ HEK-293 cells were plated in a 24-well plate and cultured for 16 h. The cells were subsequently incubated for 6 h with 50 μL of a 3:1 PEI: DNA mixture (w/w) containing 0.32 μg (total plasmid amount for FISC system) of plasmid DNA. At 24 h after transfection, cells were illuminated for different time (0 to 120 h) or different light intensities (0–5 mW cm$^{-2}$) using a custom-built $4 \times 6$ FRL LED array (730 nm, Epistar, Taiwan, China). The light intensity was determined at the wavelength of 730 nm using an optical power meter (Q8230, Advanstest, Japan), according to the manufacturer's operating instructions. SEAP expression in the culture medium was quantified at 48 h after the first illumination. EGFP expression was quantified at 48 h after the first illumination by flow cytometry.

**SEAP reporter assay**. The expression of human placental SEAP in culture supernatants was determined using a p-nitrophenylphosphate-based light absor-bance time course assay[51]. Briefly, 120 μL substrate solution [100 μL 2 × SEAP buffer containing 20 mM homoarginine, 1 mM MgCl$_2$, 21% (vol/vol) diethanola-mine, pH 9.8, and 20 mM substrate solution containing 120 mM p-nitrophenyl-phosphate] was added to 80 μL of heat-inactivated (65 °C, 30 min) cell culture supernatant. The time course of the absorbance at 405 nm was measured by the Synergy H1 hybrid multi-mode microplate reader (BioTek Instruments, Inc.) with Gen5 software (version: 2.04). Quantification of SEAP expression was calculated from the slope of the time-dependent increase in light absorbance.

**Luciferase reporter assay**. Luciferase activity levels were assayed according to the manufacturer's instructions (Beyotime, China; Cat. no. RG005). HEK-293 cells were plated at $6 \times 10^4$ cell/well in 24-well plates and transfected the following day with plasmids encoding pXY137 (P$_{hCMV}$-p65-VP64-BldD-pA::P$_{hCMV}$-BphS-P2A-YhjH-pA, 100 ng), pXY169 (P$_{hCMV}$-CreN59-L9-Coh2-NES-pA, 10 ng), pXY177 (P$_{FRLd}$-NLS-DocS-L9-CreC60-pA, 10 ng) and pXY185 (P$_{hCMV}$-loxP-STOP-loxP-Luciferase-pA, 200 ng). At 24 h after transfection, the culture plates were illumi-nated for 6 h with FRL (1.5 mW cm$^{-2}$, 730 nm) each day for 2 days. Then treated with 200 μl cell lysate, the cells were centrifuged at 13,800 × g for 5 min and the supernatants were collected. The 100 μl luciferin was added to the 100 μl super-natant, and the signal was detected using the Synergy H1 hybrid multi-mode microplate reader (BioTek Instruments, Inc.).

**Flow cytometry**. HEK-293 cells were plated at $6 \times 10^4$ cell/well in 24-well plates and transfected the following day with plasmids encoding pXY137 (100 ng), pXY169 (10 ng), pXY177 (10 ng) and pDL78 (P$_{hCMV}$-loxP-STOP-loxP-EGFP-pA). At 24 h after transfection, the culture plates were illuminated for 6 h with FRL (1.5 mW cm$^{-2}$, 730 nm) each day for 2 days. Then cells were harvested by trypsinization and washed in PBS three times. About 10,000 events were collected per sample and analyzed with a Becton Dickinson LSRFortessa$^{TM}$ Flow Cytometer (BD Bios-ciences) equipped for EGFP [488 nm laser, 505 nm longpass filter, 530/30 emission filter (passband centered on 530 nm; passband width 30 nm)] detection. Data were analysed using the FlowJo software (Version No. 7.6). DNA recombination effi-ciency mediated by FISC system was determined using weighted EGFP[52]. The transfected HEK-293 cell populations were gated for cells with high EGFP fluor-escence beyond a threshold of $10^2$ arbitrary fluorescence units. Weighted EGFP is the value of the percentage of gated cells multiplied by their median fluorescence, which is correlated with fluorescence intensity with cell number.

**Fluorescence imaging**. Fluorescence image of EGFP expression cells were performed with an inverted fluorescence microscope (Olympus IX71, TH4-200, Olympus, Japan) equipped with an Olympus digital camera (Olympus DP71, Olympus, Japan), a 495/535-nm (B/G/R) excitation/emission filter set and images were acquired with 480 nm excitation and 535 nm emission filters for EGFP signal.

**Spatial control of FRL-dependent DNA recombination in vitro**. HEK-293 cells were plated at $3.5 \times 10^6$ cells per dish into a 10-cm dish. After cultured for 18 h, the cells were transfected a 3:1 ration of PEI/DNA mixture (w/w) containing 16 µg (total plasmid amount for FISC system) of plasmid DNA. Twenty hours after the transfection, the culture dish was illuminated for 6 h each day by FRL (1.5 mW cm$^{-2}$, 730 nm) with a slit-patterned (the width of the slit: 6.5 mm) photomask made from an aluminum foil. Fluorescence images were taken at 48 h after illumination using a Clinx imaging equipment (ChemiScope 4300Pro, Clinx, Shanghai, China).

**Quantitative real-time PCR (qRT-PCR) analysis**. Tissue samples were harvested for total RNA isolation using a RNAiso Plus kit (Takara, Dalian, China; Cat. no. 9108) according to the manufacturer's instruction. A total of 500 ng RNA was reverse transcribed into cDNA using a PrimeScript RT reagent kit and treated with the gDNA Eraser (Takara, Dalian, China; Cat. no. RR047) to remove genomic DNA according to the manufacturer's protocol. Real-time PCR was performed on a Real-Time PCR Instrument (QuantStudio 3, Thermo Fisher Scientific Inc., USA) with the SYBR Premix Ex Taq kit (Takara, Dalian, China; Cat. no. RR420) to detect the target gene. The following parameters were used for the PCR: 95 °C for 10 min followed by 40 cycles at 95 °C for 30 s, 55 °C for 30 s, and 72 °C for 30 s, and a final extension at 72 °C for 10 min. The results were expressed as a relative mRNA amount using the standard ΔΔCt method. The housekeeping gene *glyceraldehyde 3-phosphate dehydrogenase* (*GAPDH*) was used as the endogenous control to normalize expression. Primer sequences are as follows: *GAPDH* sense 5'-TGTGTCCGTCGTGGATCTGA-3'; *GAPDH* antisense 5'-CCTGCTTCAC-CACCTTCTTGA-3'; *tdTomato* sense 5'-GACACCAAGCTGGACATCAC-3'; *tdTomato* antisense 5'-ACCTTGAAGCGCATGAACTC-3'.

**Western blot analysis**. Tissue samples were lysed in RIPA buffer (50 mM Tris–HCl, pH 7.5, 150 mM NaCl, 0.1% sodium deoxycholate, 0.1% SDS, 1 mM EDTA pH 8.0, 1% NP-40) containing 1 mM phenylmethanesulfonyl fluoride (PMSF). The mixture was grinded into homogenate on ice and the supernatant was collected by centrifuging at $16,200 \times g$ for 15 min at 4 °C. The concentration of the protein in samples was determined using a bicinchoninic acid assay kit (Beyotime, China; Cat. no. P0012S). Lysates were mixed with loading buffer and boiled for 10 min. Equal amount of proteins (30 µg) were run on a 10% sodium dodecyl sulfate-polyacrylamide gel (SDS–PAGE) and then transferred onto a polyvinylidene fluoride (PVDF) membrane (Millipore MA, USA; Cat. no. IPVH00010). The membrane was blocked with 5% nonfat milk in TBST buffer (50 mM Tris, 1.37 mM NaCl, 2.7 mM KCl, 0.05% Tween 20, pH 8.0) for 1 h at room temperature. The membranes were then incubated with anti-tdTomato primary antibody (1:500, GenScript, China; Cat. no. A00682) or anti-GAPDH primary antibody (1:1000, Beyotime, China; Cat. no. AF1186) overnight at 4 °C. After washing three times with TBST buffer, the membrane was then incubated with secondary antibody (Alexa Fluor790 Goat Anti-Rabbit, 1:5000 dilution, Jackson ImmunoResearch) for 1 h at room temperature. After washing three times with TBST buffer, the membrane was visualized using a fluorescent Western blot imaging system (LI-COR Odyssey Clx, USA).

**AAV production and titration**. AAV (serotype 2/8) was produced for gene delivery in vivo[53]. Briefly, packaging plasmid (pRC-2/8), purified helper plasmid (pHelper), and transfer plasmid (containing the interest cassette) were mixed at a 1:1:1 (w/w/w) ratio and transfected into HEK-293FT cells plated on a 15-cm dish ($8 \times 10^6$ cells/dish) using PEI. Seventy-two hours after transfection, cells were harvested and resuspended in 3 ml lysis buffer (20 mM Tris-HCl pH 8.0, 150 mM NaCl), following by freezing and thawing three times. After addition of MgCl$_2$ (final concentration, 1 mM) and benzonase (final concentration, 50 U ml$^{-1}$), cell lysis mixtures were incubated at 37 °C for 40 min and centrifuged at $4000 \times g$ for 10 min to collect the supernatant. The supernatant was loaded onto an iodixanol gradient and centrifuged at $350,000 \times g$ for 2 h at 4 °C. The 40% iodixanol fraction was isolated, washed with PBS and concentrated to a final volume 100–200 µL using Amicon Ultra 15 centrifugal filter devices-100 K (Millipore, Bedford, MA). The purified AAV was titered using a quantitative PCR.

**DNA recombination of the FISC system in BALB/c mice**. The male BALB/c wild-type mice (6-week-old, ECNU Laboratory Animal Center) were randomly divided into eight groups. The mice were hydrodynamically injected with plasmid DNA encoding the FISC system [pXY137 (P$_{hCMV}$-p65-VP64-BldD-pA::P$_{hCMV}$-BphS-P2A-YhjH-pA, 150 µg), pXY237 (pA-CreC60-L9-DocS-NLS-P$_{FRLd}$-Space3-P$_{hCMV}$-CreN59-L9-Coh2-NES-P2A-ZeoR-pA, 100 µg), pXY185 (P$_{hCMV}$-*loxP*-STOP-*loxP*-Luciferase-pA, 100 µg)] or the blue light-controlled CRY2-Cre system [pXY240 (P$_{hCMV}$-CIB1-L12-CreC106-IRES-CRY2-L12-CreN104-pA, 74 µg), pXY185, 100 µg] or the PA-Cre system (PA-Cre plasmid, 67 µg; pXY185, 100 µg) with the same molar concentration of Cre through tail vein or only the luciferase

reporter plasmid pXY185 (100 µg) as control. The injection volume of the DNA mixture solution was 100 µL per mouse weight (g). After 8 h injection, mice were illuminated with a far-red light physiotherapy lamp (E27, Phillips) or blue LEDs for 12 h (20 mW cm$^{-2}$; 15 min on, 15 min off, alternating) and then each mouse was intraperitoneally injected with 100 mM luciferin substrate solution (SYNCHEM; CAS NO: 115144-35-9). Three minutes after the luciferin injection, bioluminescence images of the mice were obtained using IVIS Lumina II in vivo imaging system (Perkin Elmer, USA). Radiance (p sec$^{-1}$ cm$^{-2}$ sr$^{-1}$) values were calculated for region of interest (ROI) using Living Image® 4.3.1 software.

**DNA recombination of the FISC system in the transgenic mice**. The transgenetic Cre-tdTomato reporter mice (6-week-old, male, *Gt(ROSA)26Sor$^{tm14(CAG-tdTomato)Hze}$*, from ECNU Laboratory Animal Center) were randomly divided into six groups with three mice in each group. These mice were hydrodynamically injected with plasmid DNA encoding the FISC system [pXY137 (P$_{hCMV}$-p65-VP64-BldD-pA::P$_{hCMV}$-BphS-P2A-YhjH-pA, 150 µg) and pXY237 (pA-CreC60-L9-DocS-NLS-P$_{FRLd}$-Space3-P$_{hCMV}$-CreN59-L9-Coh2-NES-P2A-ZeoR-pA, 100 µg)] or the blue light-controlled CRY2-Cre system [pXY240 (P$_{hCMV}$-CIB1-L12-CreC106-IRES-CRY2- L12-CreN104-pA, 74 µg)] or the PA-Cre system (PA-Cre, 67 µg) with the same molar concentration of Cre through tail vein. The injection volume of the DNA mixture solution was 100 µL per mouse weight (g). After 8 h injection, mice were illuminated with a far-red light physiotherapy lamp (E27, Phillips) or blue LEDs for 12 h (20 mW cm$^{-2}$; 15 min on, 15 min off, alternating). Control mice were kept in the dark. Five days after illumination, the mice were anaesthetized with isoflurane and sacrificed and the livers were isolated for fluorescence imaging or histological analysis. The tdTomato signal from isolated liver was detected using IVIS Lumina II in vivo imaging system equipped with tdTomato filter sets (Perkin Elmer, USA). The collected fluorescence emission signals were stored in epi-fluorescence units (average counts) and the total counts were calculated for ROI. All images were analyzed using the Living Image® 4.3.1 software.

For plasmid electroporation into the mouse muscles, the *Gt(ROSA)26Sor$^{tm14(CAG-tdTomato)Hze}$* mice (male, 6-week-old) were randomly divided into six groups with five mice in each group. The tibialis posterior muscles of the mice were surgically exposed and electroporated with a total of 40 µg of plasmids for the FISC system (pXY137, 20 µg; pXY237, 20 µg) or the blue light-controlled CRY2-Cre system (pXY240, 40 µg) or the PA-Cre system (PA-Cre, 40 µg). The muscle was electroporated using the TERESA-EPT-I Drug Delivery Device (Shanghai Teresa Healthcare Sci-Tech Co., Ltd., Shanghai, China) with the following parameters: 60 V (voltage), 50 ms (pulse width), 1 HZ (frequency). Twenty-four hours after electroporation, the mice were illuminated by FRL or blue light for 12 h (20 mW cm$^{-2}$; 15 min on, 15 min off, alternating) each day for 2 days, while the control mice were kept in the dark. At 48 h after the first illumination, the mice of each group were sacrificed and the muscular tissues were collected for qRT-PCR and Western blot analysis.

For the AAV-mediated FISC system experiment, the *Gt(ROSA)26Sor$^{tm14(CAG-tdTomato)Hze}$* mice (male, 8-week-old) were randomly divided into two groups with three mice in each group. These mice were transduced with a cocktail of 1 ml AAV virus mixture containing three AAV vectors: AAV-P$_{hCMV}$-BphS-pA ($2 \times 10^{11}$ vg), AAV-P$_{hCMV}$-CreN59-L9-Coh2-pA ($1 \times 10^{11}$ vg) and AAV-P$_{hCMV}$-FRTA-pA-P$_{FRLd}$-Docs-L9-CreC60-pA ($2 \times 10^{11}$ vg). Following two weeks of injection, the mice were illuminated with far-red light at intensity of 20 mW cm$^{-2}$ for 12 h (15 min on, 15 min off, alternating) each day for 2 days. At 7 days after the first illumination, the tdTomato signal from isolated liver was detected using IVIS Lumina II in vivo imaging system (Perkin Elmer, USA). Then the mice were sacrificed and the livers were isolated for qRT-PCR, Western blot, and histological analysis.

**Liver histology imaging**. Fresh livers were washed three times with cold PBS to remove impurities such as blood and then fixed in 4% (w/v) paraformaldehyde (PFA, Sangon Biotech; Cat. no. 30525-89-4) for 2 h at 4 °C. Tissue blocks of ~1 cm$^3$ were cut and embedded in opti-mum cutting temperature compound (OCT, Leica; Lot. no. 03803389). Five µm thick liver sections were prepared using Cryostat Microtome (Leica; CM1950, Clinical Cryostat) and rinsed with PBS. Finally, samples were counterstained with 4′,6-diamidino-2-phenylindole (DAPI, Sigma, CAS. 28718-90-3) for 10 min. Endogenous gene *tdTomato* expression was observed on an inverted fluorescence microscope (Leica DMI8, Wetzlar, Germany).

**Ethics**. All experiments involving animals were performed according to the protocol approved by the ECNU Animal Care and Use Committee and in direct accordance with the Ministry of Science and Technology of the People's Republic of China on Animal Care Guidelines. The protocol involved in this study was approved by the ECNU Animal Care and Use Committee (protocol ID: m20171203).

**Statistical analysis**. All data are expressed as the mean ± SD of three independent experiments. For the animal experiments, each treatment group consisted of randomly selected mice ($n = 3$–5). Statistical significance was determined by Student's *t* test (two tailed) unless otherwise noted. All curve fitting was performed with

Prism 5 software (version 5.01, GraphPad Software Inc.). The results were considered significant at $P < 0.05$ (*), very significant at $P < 0.01$ (**), and extremely significant at $P < 0.001$ (***).

**Reporting summary**. Further information on research design is available in the Nature Research Reporting Summary linked to this article.

## Data availability

All data associated with this study are present in the paper or the Supplementary Information. The source data and scanned film images for figures are provided as Source Data files. All genetic components related to this paper are available with a material transfer agreement and can be requested from H.Y. (hfye@bio.ecnu.edu.cn). Any other relevant data are available from the authors upon reasonable request. Source data are provided with this paper.

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

## Acknowledgements

This work was financially supported by the grants from the National Key R&D Program of China, Synthetic Biology Research (no. 2019YFA0904500), the National Natural Science Foundation of China (NSFC: no. 31971346, no.31861143016), the National Key R&D Program of China, Stem Cell and Translational Research (no. 2016YFA0100300), the Science and Technology Commission of Shanghai Municipality (No. 18JC1411000), the Thousand Youth Talents Plan of China and the Fundamental Research Funds for the Central Universities to H.Y.. This work was also partially supported by National Key R&D Program of China (no. 2019YFA0110802), NSFC: no. 31870861 to M.W. We also thank the ECNU Multifunctional Platform for Innovation (011) for supporting the mouse experiments and the Instruments Sharing Platform of School of Life Sciences, East China Normal University.

## Author contributions

H.Y. conceived this study. H.Y., J.W., X.Y., and M.W. designed the project. J.W., M.W., X.Y., C.Y., J.J., and Y.Y. performed the experimental work. H.Y., J.W., X.Y., and M.W. analysed the results and wrote the manuscript. All authors edited and approved the manuscript.

## Competing interests

The authors declare no competing interests.
