## [Peer Review File · Nature Communications]

Reviewers' Comments:

Reviewer #1:

Remarks to the Author:

This paper describes the development of a far red light inducible Cre-lox system in diverse mammalian cells and mice. Light is an attractive modulator of Cre activity in mice due to its relative non-invasiveness and simplicity, and its super spatial, temporal, and quantitative control properties. Far red light is particularly desirable due to its deep tissue penetration and low phototoxicity. In previous work, groups have developed UV and blue light inducible Cre-lox systems. However those systems have limitations including phototoxicity and poor tissue penetration of the controlling light wavelength. Thus, the current work is interesting to a broad audience.

To achieve this goal, the authors express a previously developed N-terminal fragment of a split Cre recombinase (CreN) fused to the Coh2 domain, which heterodimerizes with the DocS domain. They then express a DocS-fusion to the corresponding C-terminal domain of Cre (CreC) under control of their previous Bacteriophytochrome-based far red light inducible mammalian promoter system. Far red light exposure results in expression of DocS-CreC, which complements CreN-Coh2 in order to activate Cre recombinase activity, and thereby excision of DNA residing between two loxP sites. The authors demonstrate their new system can be delivered with adeno-associated virus (AAV), and can be activated with a far red LED, which should make it widely useful as an in vivo research tool. The system outperforms the blue-light inducible alternatives in vitro. Furthermore, it has an advantage over the recent near infrared light up conversion nanoparticle methods because it is fully genetically encoded and requires no introduction of nanoparticles to the tissue site of interest in the animal.

Overall, the work is well done, the results clear, and the paper well written. I believe it is suitable for publication in Nature Communications if the authors address several minor issues.

*The 4 plasmid design that the authors use to optimize their system is described in a way that is very difficult to understand (lines 120-124). Rather than relying upon the unwieldy names of each of these plasmids, the authors should describe the important components of each of the plasmids and the role of each plasmid in the overall design. I have a similar comment for lines 175-176, where they combine the plasmids for in vivo work.

*There are a number of undefined abbreviations (e.g. pA, STOP, GOI, 3*whiG) in Figure 1. Due to the very large number of elements in these figures, the authors should define everything in the figure legend.

*Why is EGFP fluorescence reported in the unusual and non-intuitive units of "Weighted EGFP" in Figure 2a as opposed to standard units like EGFP fluorescence per cell? What weighted EGFP is, why it is used, and how it is calculated should be clearly described in the legend.

*It is not immediately clear that Cre activity induces TdTomato expression in livers of the Gt(ROSA)...mouse strain. This lack of clarity makes the results of light exposure difficult to interpret at first. The authors should explain the design of this mouse strain, and the rationale for the experiment prior to discussing their experiment and results.

*The data in Figure 4, where the authors are injected the plasmids into the mice, are weak relative to the AAV-delivery data in Figure 5. It seems that researchers will be interested in the AAV delivery mode, as it is very common, and not the free plasmid delivery mode. I think Fig. 4 is a distraction and should be moved to the supplemental materials.

*There is a red light activated AAV that has been used for optogenetic activation of gene expression in vitro (Gomez et al. ACS Nano 2016). This paper is relevant prior work and should be

cited.

*The authors should move descriptions of how their system works from the Discussion to the Introduction. The Discussion should focus on the benefits and limitations of the system, future work that can be done to better understand how well it works (e.g. single cell-level, different tissues, organs, etc.), improve upon it, and future applications of the system.

Reviewer #2:

Remarks to the Author:

In this report, the authors describe a far-red light-induced split Cre (FISC) system. Far-red light (FRL) stimulates the system allowing for expression and reconstitution of a split Cre recombinase under the control of a previously described FRL inducible system. The transcriptional system works in response to FRL through generation of c-di-GMP through a bacterial phytochrome Bph and subsequent activation of a c-di-GMP responsive transcriptional activator (P65-VP64-BldD). Activity of a split-Cre (split along previously identified residues and reconstituted through the Coh2-DocS interaction) is controlled by putting the CreC component under the FRL transcriptional system promoter (PFRLx). The authors optimize many aspects of the system (PFRL promoters, linkers, lox sites) in HEK cells to allow for low background (dark) recombination and low phototoxicity, with significant recombination in FRL. Red-light is able to penetrate the organs of mice, allowing for Cre-recombinase mediated DNA recombination in the livers of BALB/c wild-type mice and in a CRE-tdTomato reporter mouse model. They can split their system between several vectors, allowing for packaging and delivery with adeno-associated virus (AAV).

Site-specific DNA recombination using Cre recombinase (which catalyzes homologous DNA recombination between specific 34-bp loxP sites) is a commonly used genome engineering technique. Certainly, an FRL light controlled Cre is a very useful alternative to chemically inducible systems (which lack spatial resolution) and blue/UV-light inducible systems (which lack depth penetration and cause phototoxicity). Overall this article is well written and the data well presented. I have a few comments that should be addressed before publication:

Major comments:

- The results in Supplemental Figure 1 are confusing to me. Unless I am reading the figure incorrectly, it shows that background activity of the split-Cre is actually higher in the dark than in the light. How is this possible? I have a similar question about Supplemental Figure 2a, b. How is it that the activity is consistently higher in the dark, across multiple transfection conditions, for PFRLAa and PFRLc?
- I don't have a perfect suggestion for how to fix this, but in places the paper is hard to follow simply because of the number of multi-component labels to follow (e.g. lines 121-124). I don't know if there is a way to shorten the "code name" for some of the components, e.g. just use CreC? Again, I don't have a perfect suggestion on how to fix this but there were a lot of construct names specific to this paper that at times became a bit unwieldy.
- Is background induction at all a function of time? For example, in Figure 2d we see that the SEAP production increases and the fold-change, but not the background induction. The data in Figure 1e (and Supplemental Figures 1,2) is taken after a specific amount of time so I don't have a sense for how it varies with time or if it continues increasing (which I would expect).
- What isn't clear to me, but should be known from experiments like the experiment depicted in Figure 2a, is what the fraction of cells completing recombination is. This seems like a relevant piece of information unless the heterogeneity of recombination is not important in downstream applications. Figure 2f does make it appear that there is some heterogeneity, though whether this is due to light leakage from the mask edges isn't obvious. It is also interesting to ask how recombination varies as a function of light intensity. Is it bimodal? I.e. at intermediate light intensities does some cells switch while others do not? Or left in intermediate light long enough would all cells switch, and so it is really the time to switching that is relevant?

- What is also not clear to me is how long the light needs to be on for maximum recombination. In Figure 2d SEAP production is measured, but I assume this just keeps increasing whether or not recombination continues happening. It might be interesting to ask how different light durations affect SEAP production (i.e. at what point do you saturate the efficacy of the system with light?)
 - Line 174—You mention optimizing the distance between the two split Cre sequences, but Supp Figure 8 shows only two constructs. Did you do extensive optimization, and did you find any trends? Or was trying one other spacing enough to dramatically improve performance? Looking through Supp Table 3 I don't see those intermediate plasmids?
 - Can you explain why your results for the PA-Cre differ from the results of Kawano, et al? They showed more induction with the PA-Cre system in vivo.
 - Should I take Figure 3C to mean there is substantial background in the mice? I don't know what the background measurement is in mice with a luciferase reporter plasmid for example.
 - Line 245, 246---How do you conclude that the AAV is more efficient than other delivery methods? What delivery methods are you directly comparing to and can you make a clear comparison figure?
 - Please compare/contrast your method to methods like Kinjo, et al 2019 (Nature Methods) which allow for two-photon (2P) excitation by developing a FRET transfer from BFP to CRY2.
- Minor comments:
- Line 73—"of UCNPs"
 - Line 101—FRTA acronym introduced without explanation. I had to go to Shao, et al to know that it refers to synthetic mammalian transactivators.
 - Is there a justification/explanation for why only inducing expression of one of the split-Cre components is preferable (particularly in terms of background) to inducing both of them? This doesn't need to be tested experimentally but could be minimally discussed as part of an explanation for the design or in the discussion.
 - In a similar vein, is there any intuition for why the different linkers were tested and why they worked more or less well?
 - The explanation of the promoter construction (mostly given in Figure 1c without a detailed caption) could be expanded in the main text to explain the promoter structure.
 - Can you discuss why there is such a range of efficiency between cell lines, and does this mean tissue specific/cell line optimization would be necessary for this technology to be adopted by other labs?
 - Supplemental Figure 6—please double check that the 72h FISC points are not identical—the three points look identical to my eyes between the dark and illuminated samples, but it could just be my eyesight.
 - Line 195—I think there is an "it" missing before "superior"
 - Please clarify in the methods what the epi-fluorescence units are (counts/total counts) and how they were measured.
 - It is unfortunately that the spatial resolution of light is not employed in this paper in vivo. That would be exciting.
 - Line 272—I think potentially two sentences were accidentally concatenated here. The sentence as is doesn't make sense.

Manuscript NCOMMS-20-02225-T—Point-by-point responses to referees' comments:

Title: "A non-invasive far-red light-induced split-Cre recombinase system for controllable genome engineering in mice"

We would like to thank all of the referees for their highly constructive comments. As we hope you will agree, the careful revision process we have undertaken has substantially improved both the scientific rigor and impact of our study. We present point-by-point responses to each of the referee comments (below). We therefore invite you to examine our responses below, and we would again like to thank all of the referees in the editor for their ongoing work on our behalf.

Reviewer #1 (Remarks to the Author):

This paper describes the development of a far red light inducible Cre-lox system in diverse mammalian cells and mice. Light is an attractive modulator of Cre activity in mice due to its relative non-invasiveness and simplicity, and its super spatial, temporal, and quantitative control properties. Far red light is particularly desirable due to its deep tissue penetration and low phototoxicity. In previous work, groups have developed UV and blue light inducible Cre-lox systems. However those systems have limitations including phototoxicity and poor tissue penetration of the controlling light wavelength. Thus, the current work is interesting to a broad audience.

To achieve this goal, the authors express a previously developed N-terminal fragment of a split Cre recombinase (CreN) fused to the Coh2 domain, which heterodimerizes with the DocS domain. They then express a DocS-fusion to the corresponding C-terminal domain of Cre (CreC) under control of their previous Bacteriophytochrome-based far red light inducible mammalian promoter system. Far red light exposure results in expression of DocS-CreC, which complements CreN-Coh2 in order to activate Cre recombinase activity, and thereby excision of DNA residing between two loxP sites. The authors demonstrate their new system can be delivered with adeno-associated virus (AAV), and can be activated with a far red LED, which should make it widely useful as an in vivo research tool. The system outperforms the blue-light inducible alternatives in vitro. Furthermore, it has an advantage over the recent near infrared light up conversion nanoparticle methods because it is fully genetically encoded and requires no introduction of nanoparticles to the tissue site of interest in the animal.

Overall, the work is well done, the results clear, and the paper well written. I believe it is suitable for publication in Nature Communications if the authors address several minor issues.

We are excited about this enthusiastic comment and the following constructive suggestions.

-The 4 plasmid design that the authors use to optimize their system is described in a way that is very difficult to understand (lines 120-124). Rather than relying upon the unwieldy names of each of these plasmids, the authors should describe the important components of each of the plasmids and the role of each plasmid in the overall design. I have a similar comment for lines 175-176, where they combine the plasmids for in vivo work.

Thanks for this suggestion. We have made corrections accordingly in the revised manuscript. Please see Lines 125-128, 183, 188-189.

-There are a number of undefined abbreviations (e.g. pA, STOP, GOI, 3*whiG) in Figure 1. Due to the very large number of elements in these figures, the authors should define everything in the figure legend.

Thanks for this helpful suggestion. We have now defined these abbreviations in the relevant figure legend (**Fig. 1a-c**). Please see Page 24-25, and Lines 670-672, 677-685.

-Why is EGFP fluorescence reported in the unusual and non-intuitive units of “Weighted EGFP” in Figure 2a as opposed to standard units like EGFP fluorescence per cell? What weighted EGFP is, why it is used, and how it is calculated should be clearly described in the legend.

Sorry for not having described this more clearly. Weighted EGFP is the value of the percentage of gated cells multiplied by their median fluorescence, which is correlated of fluorescence intensity with cell number. This method was widely reported in the literature (Ausländer D, *et al.*, *Nature*, 2012; Yin, J., *et al.*, *Science Translational Medicine*, 2019). We have provided the detail information in the figure legend and the Method section in the revised manuscript. Please see Page 18, Lines 520-525, and Page 26, Line 713.

-It is not immediately clear that Cre activity induces TdTomato expression in livers of the Gt(ROSA)...mouse strain. This lack of clarity makes the results of light exposure difficult to interpret at first. The authors should explain the design of this mouse strain, and the rationale for the experiment prior to discussing their experiment and results.

Thanks for the constructive comment. We have now provided the detailed description about the rationale for the experiment in the revised manuscript. Please see Lines 209-212. We also have provided a **New Fig. 4b** to show schematic illustration of the working principle for transgenic Cre-tdTomato reporter mice in which the floxed-STOP cassette is excised by Cre recombinase to allow Cre reporter (tdTomato) expression. Please see Page 29, Lines 751-753.

-The data in Figure 4, where the authors are injected the plasmids into the mice, are weak relative to the AAV-delivery data in Figure 5. It seems that researchers will be interested in the AAV delivery mode, as it is very common, and not the free plasmid delivery mode. I think Fig. 4 is a distraction and should be moved to the supplemental materials.

Thanks for the suggestion. We agree with you. We have moved the old Figure 4 to the supplemental materials in the revised manuscript. Please see **new Supplementary Figure 11-12**.

-There is a red light activated AAV that has been used for optogenetic activation of gene expression in vitro (Gomez et al. ACS Nano 2016). This paper is relevant prior work and should be cited.

We have now cited this study (Reference 42) in our revised manuscript. Please see Lines 396-398.

-The authors should move descriptions of how their system works from the Discussion to the Introduction. The Discussion should focus on the benefits and limitations of the system, future work that can be done to better understand how well it works (e.g. single cell-level, different tissues, organs, etc.), improve upon it, and future applications of the system.

Thanks for the helpful suggestion. The description of the system in the Discussion part was removed. We have discussed the benefits and limitations of the system in the revised manuscript. Please see Page 10-11.

Reviewer #2 (Remarks to the Author):

In this report, the authors describe a far-red light-induced split Cre (FISC) system. Far-red light (FRL) stimulates the system allowing for expression and reconstitution of a split Cre recombinase under the control of a previously described FRL inducible system. The transcriptional system works in response to FRL through generation of c-di-GMP through a bacterial phytochrome Bph and subsequent activation of a c-di-GMP responsive transcriptional activator (P65-VP64-BldD). Activity of a split-Cre (split along previously identified residues and reconstituted through the Coh2-DocS interaction) is controlled by putting the CreC component under the FRL transcriptional system promoter (PFRLx). The authors optimize many aspects of the system (PFRL promoters, linkers, lox sites) in HEK cells to allow for low background (dark) recombination and low phototoxicity, with significant recombination in FRL. Red-light is able to penetrate the organs of mice, allowing for Cre-recombinase mediated DNA recombination in the livers of BALB/c wild-type mice and in a CRE-tdTomato reporter mouse model. They can split their system between several vectors, allowing for packaging and delivery with adeno-associated virus (AAV).

Site-specific DNA recombination using Cre recombinase (which catalyzes homologous DNA recombination between specific 34-bp loxP sites) is a commonly used genome engineering technique. Certainly, an FRL light controlled Cre is a very useful alternative to chemically inducible systems (which lack spatial resolution) and blue/UV-light inducible systems (which lack depth penetration and cause phototoxicity). Overall this article is well written and the data well presented. I have a few comments that should be addressed before publication:

We are grateful for the enthusiastic comments and the constructive suggestions.

Major comments:

- The results in Supplemental Figure 1 are confusing to me. Unless I am reading the figure incorrectly, it shows that background activity of the split-Cre is actually higher in the dark than in the

light. How is this possible? I have a similar question about Supplemental Figure 2a, b. How is it that the activity is consistently higher in the dark, across multiple transfection conditions, for PFRLAa and PFRLc?

Thanks for noting this. At the beginning of this work, we examined the recombination efficiency mediated by the CreN59-Coh2 constitutively driven by the promoter P_{hCMV} and the DocS-CreC60 expression induced by the FRL-responsive promoter P_{FRLa} , and our initial result showed that this combination had strong background activity in the dark condition (**Supplementary Figure. 1**). We think there are mainly two possible reasons that the background activity of the initial version of the split-Cre is higher in the dark than even in the light. First, the relative high leakiness expression of DocS-CreC60 driven by the promoter P_{FRLa} . Even in the dark, the leakiness expression of DocS-CreC60 can sufficiently restore split-Cre recombinase catalytic activity resulting in high background leakage in the absence of light. Second, according to the literature, the overexpression of Cre recombinase can cause the growth-inhibitory and genotoxic effects in cultured cells, reported by Ate Loonstra *et al.* (Growth inhibition and DNA damage induced by Cre recombinase in mammalian cells. *PNAS*, 2001). Therefore, the strong inducible expression of DocS-CreC60 driven by P_{FRLa} under illumination caused the strong split-Cre recombinase catalytic activity that may affect the proliferation of cells resulting in less SEAP expression in the light than that in the dark.

Subsequently, we therefore focused our efforts on reducing FRL-induced DocS-CreC60 expression driven by different promoter configurations (e.g., P_{FRLa} , P_{FRLc} , and P_{FRLd}), and transfected with different amounts of the plasmid (**Supplementary Figure 2a-c**). Among multiple tests, we found that low background expression of the reporter SEAP in the dark and relative high induction level in the light was observed under the control of a minimal eukaryotic promoter with only TATA box (P_{FRLd}) (**Supplementary Figure 2c**).

- I don't have a perfect suggestion for how to fix this, but in places the paper is hard to follow simply because of the number of multi-component labels to follow (e.g. lines 121-124). I don't know if there is a way to shorten the "code name" for some of the components, e.g. just use CreC? Again, I don't have a perfect suggestion on how to fix this but there were a lot of construct names specific to this paper that at times became a bit unwieldy.

Thanks for this comment. This is the same question also raised by the first reviewer. We have made corrections accordingly in the revised manuscript. Please see Page 5, Lines 125-128.

- Is background induction at all a function of time? For example, in Figure 2d we see that the SEAP production increases and the fold-change, but not the background induction. The data in Figure 1e (and Supplemental Figures 1,2) is taken after a specific amount of time so I don't have a sense for how it varies with time or if it continues increasing (which I would expect).

Thanks for the comment. Actually, in **new Figure 2d**, the cells transfected with the FISC system without illumination (indicated as “0”) showed very weak background of the FISC system at 120 hours after the first illumination.

Any way, we also have performed new experiments to show the time-dependent SEAP production mediated by the FISC system (**new Supplementary Figure 6**). Our new data showed that there is no significant increasing of background induction from 36-120 hours, but with significant time-dependent increasing of SEAP expression mediated by the FISC system under the far-red-light illumination.

- What isn't clear to me, but should be known from experiments like the experiment depicted in Figure 2a, is what the fraction of cells completing recombination is. This seems like a relevant piece of information unless the heterogeneity of recombination is not important in downstream applications. Figure 2f does make it appear that there is some heterogeneity, though whether this is due to light leakage from the mask edges isn't obvious. It is also interesting to ask how recombination varies as a function of light intensity. Is it bimodal? I.e. at intermediate light intensities does some cells switch while others do not? Or left in intermediate light long enough would all cells switch, and so it is really the time to switching that is relevant?

Thanks for your questions and comments. We have performed new experiments to calculate the recombination efficiency mediated by the FISC system. According to our new flow cytometric data, the fraction of the recombination is around 23.5% (**new Supplementary Figure 5a, b**).

As for the heterogeneity of recombination of the FISC system in cells, we think this is mainly due to the transfection efficiency. Because we cannot achieve 100% of the cells co-transfected with the four-plasmid mixture and with equal plasmid molecules in each cell. Normally, we can achieve 80-90% transfection efficiency of HEK-293 cells with PEI transfection method. That's why it seems that there is some heterogeneity of the cell population in Figure 2f. This phenomenon is widely existed in other reported literatures (Shao *et al.*, *PNAS*, 2018; LR Polstein, *et al.*, *Nature Chemical Biology*, 2015; K Müller, *et al.*, *Nature Protocol*, 2014; Fuun Kawano, *et al.*, *Nature Chemical Biology*, 2016).

We now have provided the flow cytometry data in **new Supplementary Figure 5a**. As we can see from the flow cytometric contour plots, some cells present high levels of EGFP expression, and some cells present lower EGFP expression, which also indicted the heterogeneity in different single cells.

The light-induced recombination efficiency is both illumination intensity- and illumination time-

dependent, as shown in **Figure 2c** and **d**. We think it is illumination time and illumination intensity relevant to activate the FISC system.

- What is also not clear to me is how long the light needs to be on for maximum recombination. In Figure 2d SEAP production is measured, but I assume this just keeps increasing whether or not recombination continues happening. It might be interesting to ask how different light durations affect SEAP production (i.e. at what point do you saturate the efficacy of the system with light?)

Thanks for this comment. We have performed additional new experiments to address this concern. Our new experimental data showed that the DNA recombination efficiency was illumination time- and intensity-dependent, and achieved a constant level (or saturation) in the following 96 h or illumination intensity above 4 mW/cm² (**revised new Fig. 2c, d**).

- Line 174—You mention optimizing the distance between the two split Cre sequences, but Supp Figure 8 shows only two constructs. Did you do extensive optimization, and did you find any trends? Or was trying one other spacing enough to dramatically improve performance? Looking through Supp Table 3 I don't see those intermediate plasmids?

Thanks for this comment. In fact, at the beginning, we tested different insulators between promoters for the two split-Cre sequences. But we did not observe substantial improvement of the FRL induction. And then we tested different distance between two promoters and found that one of the distances (Space 3) yielded the highest fold induction (18.5-fold induction) upon FRL illumination. We now have provided the detailed optimization data in the revised manuscript (**new Supplementary Figure 10**). We also have provided the related plasmids used in the **revised Supplementary Table 4**.

- Can you explain why your results for the PA-Cre differ from the results of Kawano, et al? They showed more induction with the PA-Cre system in vivo.

Thanks for the question. The main reason is that we used different experimental parameters (different amount of the plasmids and the different illumination conditions). In the PA-Cre system developed by Kawano et. al., each mouse was injected with a total amount of 300 µg plasmids encoding PA-Cre and the reporter Floxed-STOP Fluc at a 1:9 ratio, and the mice were illuminated with blue light (470±20 nm; 20 mW/cm²) for continuous 16 h or 30 s.

In order to compare the DNA recombination efficiency mediated by the two systems, the mice were injected with the same molar concentration of split-Cre. In our study, each mouse was injected with a total amount of 167 µg PA-Cre system (PA-Cre plasmid: 67 µg, the reporter pXY185, 100 µg) and illuminated with blue light (460 nm±20 nm, 20 mW/cm²,) for 12 h (15 min on, 15 min off,

alternating).

- Should I take Figure 3C to mean there is substantial background in the mice? I don't know what the background measurement is in mice with a luciferase reporter plasmid for example.

Thanks for this comment. We have performed a new control experiment that control mice were only hydrodynamically injected with the reporter plasmid (pXY185, P_{hCMV}-*loxP*-STOP-*loxP*-Luciferase-pA). The data showed that the background in the control mice was similar to that in mice treated with FISC system in the dark (**revised Fig. 3b, c**), indicating no substantial background of the FISC system.

In addition, our new *in vitro* experiment data (**new Supplementary Figure 6**) also showed the very weak background of the FISC system.

- Line 245, 246---How do you conclude that the AAV is more efficient than other delivery methods? What delivery methods are you directly comparing to and can you make a clear comparison figure?

Thanks for this comment. In this work, we have studied the DNA recombination efficiency mediated by our FISC system using three different delivery methods (AAV, hydrodynamic injection, and electroporation) in tdTomato transgenic mice.

As shown in **Figure 4**, the isolated livers from mice transduced with AAV-delivered FISC system exhibited a **20.4-fold** increase of tdTomato signal (**Fig. 4f, g**) compared to the transduced control mice in the dark. However, the isolated livers of the FRL-illuminated mice transfected with FISC system using hydrodynamic tail vein (HTV) injection exhibited a **3.5-fold** increase of tdTomato signal compared to control mice in the dark (**Supplementary Figure 11b, c**). Besides, the qRT-PCR analysis of tdTomato expression in Cre-tdTomato reporter mice transduced with AAV-delivered FISC system exhibited a **4.7-fold** (**Fig. 4h**) increase compared to the transduced control mice in the dark, but we observed an increase in tdTomato expression (**2.9-fold, Supplementary Figure 12b**) in the leg muscles of tdTomato transgenic mice using electroporation of the FISC system compared with control mice in the dark.

Therefore, these results demonstrate that the AAV-delivered FISC system showed very more efficient performance as compared with the other two delivery methods in mice. As this reviewer suggested, we now have provided a comparison table (**new Supplementary Table 3**) in the revised manuscript, and also described more clearly in the revised manuscript. Please see Lines 258-259.

- Please compare/contrast your method to methods like Kinjo, et al 2019 (Nature Methods) which allow for two-photon (2P) excitation by developing a FRET transfer from BFP to CRY2.

Thanks for this comment. It's a fantastic work that they developed a CRY2 variant, named 2paCRY2, that is efficiently activated by two-photon (2P) excitation via FRET from blue fluorescent protein (BFP) (Kinjo, *et al.*, *Nature Methods*, 2019). They finally used a Förster resonance energy transfer (FRET)-assisted photoactivation (FRAPA) to control ERK signaling pathway at single-cell resolution in the mouse auricular epidermis. The laser power (the excitation wavelengths of 810 nm) was used to induce ERK activation by 2P excitation via FRET from blue fluorescent protein (BFP) to CRY2. However, our FISC system was developed to induce DNA recombination controlled simply by far-red light (FRL, ~730 nm) from an LED.

In our opinion, they are two different tools for two different purposes. One is developed for controlling protein interaction by 2P excitation, the other is developed for controlling DNA recombination by FRL. However, there is the possibility that the 2paCRY2 could be further developed to control Cre activity to control DNA recombination. The uncertainty is that the activation efficacy of deep internal organs (eg. livers) by 2P excitation via FRET, which also needs certain high hardware requirements for the experimental setup.

We have mentioned and cited this work in the introduction part. Please see Lines 73-75, 380-381.

Minor comments:

- Line 73—"of UCNPs"

We have revised as directed. Please see Line 71.

- Line 101—FRTA acronym introduced without explanation. I had to go to Shao, et al to know that it refers to synthetic mammalian transactivators.

Sorry for not having described this more clearly. We have provided the detail information in the revised manuscript. Please see Lines 102, 672.

- Is there a justification/explanation for why only inducing expression of one of the split-Cre components is preferable (particularly in terms of background) to inducing both of them? This doesn't need to be tested experimentally but could be minimally discussed as part of an explanation for the design or in the discussion.

Thanks for this comment. It was reported that the active site of Cre recombinase is located in the larger C-terminal domain (Feng Guo *et al.*, *Nature*, 1997). Thus, in our design, we used light-responsive promoter to drive the C-terminal Cre fragment (CreC60) expression, and the other N-terminal fragment was driven by a constitutive promoter. Actually, we also have tested to induce

both of the two fragments expression by light, but we did not observe efficient light-inducible DNA recombination efficacy. We now have mentioned this point in the revised manuscript. Please see Lines 106-107.

- In a similar vein, is there any intuition for why the different linkers were tested and why they worked more or less well?

Thanks for this comment. We think that a major steric hindrance could be induced by the fusion of Coh2 and DocS to the Cre fragments, that might impede the reconstitution of an active structure of Cre. Moreover, based on the previously described regulatable version of Cre (Jullien *et al*, *Nucleic Acids Research*, 2003; Kawano *et al*, *Nature Chemical Biology*, 2016), we therefore designed and tested different linkers between CreN59 and Coh2, as well as between CreC60 and DocS. We have tested/optimized different flexibility and length of the linker. We found that one of the combinations (L9/L9) yielded the highest fold induction (50.9-fold induction) upon FRL illumination. We have provided the detail information in the **revised Supplementary Table 1**.

We also have included the explanation in the revised manuscript. Please see Lines 131-133.

- The explanation of the promoter construction (mostly given in Figure 1c without a detailed caption) could be expanded in the main text to explain the promoter structure.

Thanks for this comment. We have revised as suggested. Please see Page 4, Lines 118-119, and Page 25, Lines 683-685.

- Can you discuss why there is such a range of efficiency between cell lines, and does this mean tissue specific/cell line optimization would be necessary for this technology to be adopted by other labs?

Thanks for this comment. We think such a range of different efficiency among different cell lines could probably be the different transfection efficiency and the potential interactions with endogenous cell components of different cell lines. This is a very common phenomenon observed in other control systems or genetic switches (Bai *et al.*, *Nature Medicine*, 2019; Yin *et al.*, *Science Translational Medicine*, 2019; Shao *et al.*, *Science Translational Medicine*, 2017; Nihongaki Y, *et al.*, *Nature Methods*, 2017; Alejandro Chavez, *et al.*, *Nature Methods*, 2016;). At this stage, we think the FISC system is ready for mammals.

We also have provided the discussion in the revised manuscript. Please see Lines 154-157.

- Supplemental Figure 6—please double check that the 72h FISC points are not identical—the three

points look identical to my eyes between the dark and illuminated samples, but it could just be my eyesight.

Thank you so much for your careful review. The data of the 72 h FISC points are not identical. We have checked the original data. Please also see below:

FISC Dark 72h	450.1326	434.2624	463.3741
FISC Illumination 72h	452.1414	431.1142	465.7062

- Line 195—I think there is an “it” missing before “superior”

Sorry for this mistake. We have corrected it in the revised manuscript. Please see Line 204.

- Please clarify in the methods what the epi-fluorescence units are (counts/total counts) and how they were measured.

Thanks for this suggestion. We have provided the information in the revised manuscript. Please see Lines 618-621.

- It is unfortunately that the spatial resolution of light is not employed in this paper *in vivo*. That would be exciting.

Thanks for the comment. This is a nice suggestion. An ongoing research project in our group is examining to use the FISC system to spatiotemporally control gene expression *in vivo*. We hope this independent story will be published in the near future. We also have discussed the potential applications of the FISC system (tissue- or even cell-type specific deployment for high resolution studies) in the discussion section of our revised manuscript. Please see Lines 285, 291-293, 304-306.

- Line 272—I think potentially two sentences were accidentally concatenated here. The sentence as is doesn't make sense.

We have corrected it with the corresponding changes in the revised manuscript. Please see Lines 288-293.

Reviewers' Comments:

Reviewer #2:

Remarks to the Author:

In this article, the authors describe a far-red light-induced split CRE (FISC) system. This is an undoubtedly useful system and the original version of the article contained compelling data. My initial review was overall enthusiastic, asking for some clarification of experimental issues, writing, and figures.

In this revision, the authors have carefully addressed my original comments. Specifically, they have:

- rewritten their description of the various constructs using labeling that will be easier for the reader to understand
- clarified, in some cases with new experiments, questions related to the level of basal/dark activation
- provided new experiments to clarify time-dependent SEAP production mediated by the FISC system
- provided new data on the recombination efficiency mediated by the FISC system
- quantified the time-dependence of recombination after light exposure
- provided further information on how their plasmid constructs were optimized and provided additional clarifying text regarding their design decisions.
- added citations to relevant related literature in the introduction

Overall, I am satisfied that the authors have addressed my concerns and overall improved their manuscript. It is appropriate for publication in Nature Communications.

Manuscript NCOMMS-20-02225A—Point-by-point responses to referees' comments and editorial requests:

Title: "A non-invasive far-red light-induced split-Cre recombinase system for controllable genome engineering in mice"

We would like to thank the editor and the referees for their highly constructive comments. My co-authors and I have very carefully considered each of the editorial requests. We present point-by-point responses to each of the requests (below). We therefore invite you to examine our responses below, and we would again like to thank the editor for ongoing work on our behalf.

REVIEWERS' COMMENTS:

Reviewer #2 (Remarks to the Author):

In this article, the authors describe a far-red light-induced split CRE (FISC) system. This is an undoubtedly useful system and the original version of the article contained compelling data. My initial review was overall enthusiastic, asking for some clarification of experimental issues, writing, and figures.

In this revision, the authors have carefully addressed my original comments. Specifically, they have:

- rewritten their description of the various constructs using labeling that will be easier for the reader to understand
- clarified, in some cases with new experiments, questions related to the level of basal/dark activation
- provided new experiments to clarify time-dependent SEAP production mediated by the FISC system
- provided new data on the recombination efficiency mediated by the FISC system
- quantified the time-dependence of recombination after light exposure
- provided further information on how their plasmid constructs were optimized and provided additional clarifying text regarding their design decisions.
- added citations to relevant related literature in the introduction

Overall, I am satisfied that the authors have addressed my concerns and overall improved their manuscript. It is appropriate for publication in Nature Communications.

We appreciate the enthusiastic evaluation of our manuscript.